# Improving Policy Learning via Language Dynamics Distillation

Victor Zhong[1,2], Jesse Mu[3], Luke Zettlemoyer[1,2], Edward Grefenstette[4,5] and Tim Rocktäschel[4]

[1]University of Washington
[2]Meta AI Research
[3]Stanford University
[4]University College London
[5]Cohere

## Abstract

Recent work has shown that augmenting environments with language descriptions improves policy learning. However, for environments with complex language abstractions, learning how to ground language to observations is difficult due to sparse, delayed rewards. We propose Language Dynamics Distillation (LDD), which pretrains a model to predict environment dynamics given demonstrations with language descriptions, and then fine-tunes these language-aware pretrained representations via reinforcement learning (RL). In this way, the model is trained to both maximize expected reward and retain knowledge about how language relates to environment dynamics. On SILG, a benchmark of five tasks with language descriptions that evaluate distinct generalization challenges on unseen environments (NetHack, ALFWorld, RTFM, Messenger, and Touchdown), LDD outperforms tabula-rasa RL, VAE pretraining, and methods that learn from unlabeled demonstrations in inverse RL and reward shaping with pretrained experts. In our analyses, we show that language descriptions in demonstrations improve sample-efficiency and generalization across environments, and that dynamics modeling with expert demonstrations is more effective than with non-experts.

## 1 Introduction

Language is a powerful medium that humans use to reason about abstractions—its compositionality allows efficient descriptions that generalize across environments and tasks. Consider an agent that follows instructions to clean the house (e.g. find the dirty dishes and wash them). In tabula-rasa reinforcement learning (RL), the agent observes raw perceptual features of the environment, then grounds these visual features to language cues to learn how to behave through trial and error. In contrast, we can provide the agent with language descriptions that describe abstractions which are present in the environment (e.g. *there is a sink to your left and dishes on a table to your right*), thereby simplifying the grounding challenge. Language descriptions of observations occur naturally in many environments such as text prompts in graphical user interfaces [Liu et al., 2018], dialogue [He et al., 2018], and interactive games [Küttler et al., 2020]. Recent work has also shown improvements in visual manipulation [Shridhar et al., 2021] and navigation [Zhong et al., 2021, Tam et al., 2022] by captioning the observations with language descriptions. Despite these gains, learning how to interpret language descriptions is difficult through RL, especially on environments with complex language abstractions [Zhong et al., 2021].

---

Corresponding author Victor Zhong `vzhong@cs.washington.edu`

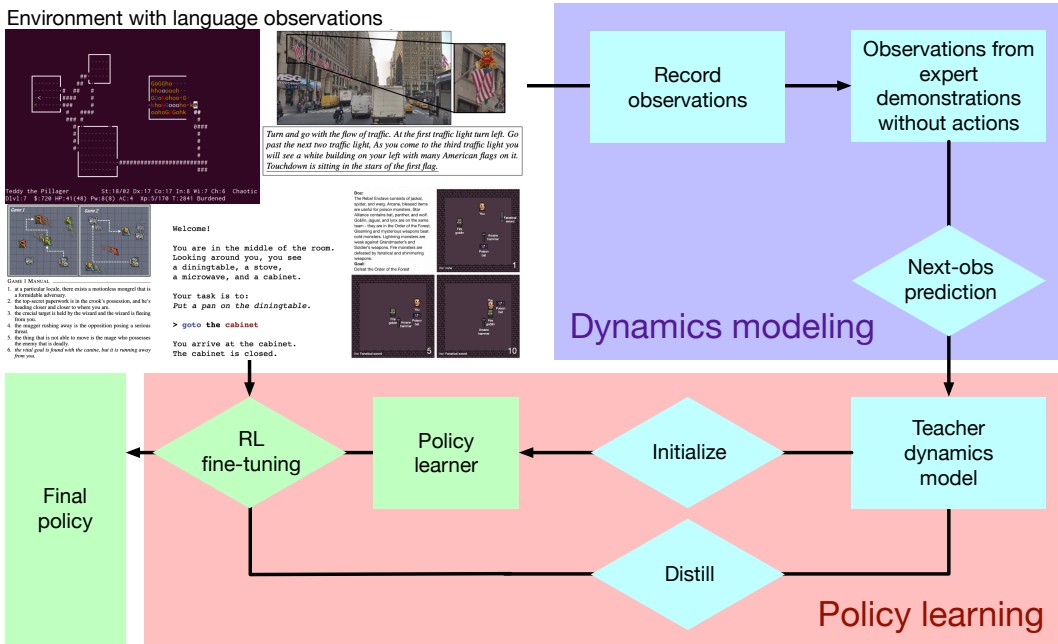

Figure 1: Language Dynamics Distillation (LDD). LDD uses cheap unlabeled demonstrations to learn a dynamics model of the environment, which is used to initialize and distill grounded representations into the policy learner. During the dynamics modeling phase (purple), we train a teacher model to predict the next observation given prior observations using unlabeled demonstrations. In the policy learning phase (red), we initialize a model with the teacher and distill intermediate representations from the teacher during reinforcement learning. The traditional policy learning loop is shown in green. LDD-specific components are shown in blue.

We present Language Dynamics Distillation (LDD), a method that improves RL by learning a dynamics model on cheaply obtained unlabeled (i.e. no action labels) demonstrations with language descriptions. When learning how to use language descriptions effectively, one central challenge is how to disentangle language understanding from policy performance from sparse, delayed rewards. Our motivation is to learn initial language grounding via dynamics modeling from an offline dataset, away from the credit assignment and non-stationarity challenges posed by RL. While labeled demonstrations that tell the agent how to to act in each situation are expensive to collect, for many environments one can cheaply obtain unlabeled demonstrations (e.g. videos of experts performing the task) [Yang et al., 2019, Stadie et al., 2017]. Intuitively, LDD exploits these unlabeled demonstrations to learn how to associate language descriptions with abstractions in the environment. This knowledge is then used to bootstrap and more quickly learn policies that generalize to instructions and manuals in new environments. Given unlabeled demonstrations with language descriptions (e.g. captions of scene content), we first pretrain the model to predict the next observation given prior observations, similar to language modeling. A copy of this model is stored as a fixed teacher that grounds language descriptions to predict environment dynamics. We then train a model with RL, while distilling intermediate representations from the teacher to avoid catastrophic forgetting of how to interpret language descriptions for dynamics modeling. In this way, the model learns to both maximize expected reward while retaining knowledge about how language descriptions relate to environment dynamics.

We evaluate LDD on the recent SILG benchmark [Zhong et al., 2021], which consists of five diverse environments with language descriptions including NetHack [Küttler et al., 2020], ALFWorld [Shridhar et al., 2021], RTFM [Zhong et al., 2020], Messenger [Hanjie et al., 2021], and Touchdown [Chen et al., 2018]. These environments present unique challenges in language-grounded policy-learning across complexity of instructions, visual observations, action space, reasoning procedure, and generalization. By learning a dynamics model from cheaply obtained unlabeled demonstrations, LDD consistently outperforms reinforcement learning with language descriptions both in terms of sample efficiency and generalization performance. Moreover, we compare LDD to other techniques that inject prior knowledge in VAE pretraining [Kingma and Welling, 2013], inverse reinforcement learning [Hanna and

Stone, 2017, Torabi et al., 2018, Guo et al., 2019], and reward shaping with a pretrained expert [Merel et al., 2017]. LDD achieves top performance on all environments in terms of task completion and reward. In addition to comparing LDD to other methods, we ablate LDD to quantify the effect of language observations in dynamics modeling, and the importance of dynamics modeling with expert demonstrations. On two environments where we can control for the presence of language descriptions (NetHack game messages and Touchdown panorama captions), we show that language descriptions improve sample-efficiency and generalization. Finally, across all environments, we find that dynamics modeling with expert demonstrations is more effective than with non-expert rollouts.

## 2  Related Work

**Learning by observing language.**    Recent work studies generalization to language instructions and manuals that specify new tasks and environments. These settings range from photorealistic/3D navigation [Anderson et al., 2018, Chen et al., 2018, Ku et al., 2020, Shridhar et al., 2020] to multi-hop reference games [Narasimhan et al., 2015, Zhong et al., 2020, Hanjie et al., 2021]. We use a collection of these tasks to evaluate LDD. There is also work where understanding language is not necessary to achieve the task, however its inclusion (e.g. via captions, scene descriptions) makes learning more efficient. Shridhar et al. [2021] show that one can quickly learn policies in a simulated kitchen environment described in text, then transfer this policy to the 3D visual environment. Zhong et al. [2021] similarly transform photorealistic navigation to a symbolic form via image segmentation, then learn a policy that transfers to the original photorealistic setting. In work concurrent to ours, Tam et al. [2022] generate oracle captions of observations for simulated robotic control and city navigation, which improve policy learning. LDD is complementary to these—in addition to incorporating language descriptions as features, we show that learning a dynamics model from unlabeled demonstrations with language descriptions improves sample efficiency and results in better policies.

**Imitation learning from observations.**    There is prior work on model-free as well as model-based imitation learning from observations. Model-free methods encourage the imitator to produce state distributions similar to those produced by the demonstrator, for example via generative adversarial learning [Merel et al., 2017] and reward shaping [Kimura et al., 2018]. In contrast, LDD only requires intermediate representations extracted from an expert dynamics model on states encountered by the learner, which are cheaper to compute than rollouts from an expert policy. Model-based approaches learn dynamics models that predict state-transitions given the current state and an action. Hanna and Stone [2017] learn an inverse model to map state-transitions to actions, which is then used to annotate unlabeled trajectories for imitation learning. Edwards et al. [2019] learn a forward dynamics model that predicts future states given state and latent action pairs. In contrast, LDD does not assume priors over the action space distribution. For instance, on ALFWorld, our method works even though it is impossible to enumerate the action space. In our experiments, we extend model-free reward shaping and model-based inverse dynamics modeling to account for language descriptions and compare LDD to these methods.

**Representation learning in RL.**    In representation learning for RL, the agent learns representations of the environment using rewards and objectives based on the difference between the state and prior states [Strehl and Littman, 2008], raw visual observations [Jaderberg et al., 2017], learned agent representations [Raileanu and Rocktäschel, 2020], and random network observations [Burda et al., 2019]. In intrinsic exploration methods [Raileanu and Rocktäschel, 2020, Burda et al., 2019], the training objective encourages dissimilarity (e.g. in observation/state space) to prior agent experience so that the agent discovers novel states. Unlike intrinsic exploration, the distillation objective in Language Dynamics Distillation encourages similarity to expert behaviour, as opposed to dissimilarity to prior agent experience. In reconstruction based representation learning methods [Strehl and Littman, 2008, Jaderberg et al., 2017], the training objective encourages the agent to learn intermediate representations that also capture the dynamics and structure of the environment by reconstructing the observations (e.g. predicting what objects are in scene). Language Dynamics Distillation is similar to reconstruction methods for representation learning, however unlike the latter, the dynamics model in LDD is trained on trajectories obtained from an expert policy as opposed to the agent policy. Language Dynamics Distillation is complementary to intrinsic exploration methods and to reconstruction based representation learning methods.

# 3 Language Dynamics Distillation

Recent work improves policy learning by augmenting environment observations with language descriptions [Shridhar et al., 2021, Zhong et al., 2021, Tam et al., 2022]. For environments with complex language abstractions, however, learning how to associate language to environment observations is difficult through RL due to sparse, delayed rewards. In Language Dynamics Distillation (LDD), we pretrain the model on unlabeled demonstrations (i.e. no annotated actions) with language descriptions to predict the dynamics of the environment, then fine-tune the language-aware model via RL. LDD consists of two phases. In the first dynamics modeling phase, we pretrain the model to predict future observations given unannotated demonstrations. We store a copy of the model as a fixed teacher that has learned grounded representations useful for predicting how the environment behaves under an expert policy. In the second reinforcement learning phase, we fine-tune the model through policy learning, while distilling representations from the teacher. This way, the model is trained to both maximize expected reward and retain knowledge about the dynamics of the environment. Fig 1 illustrates the components of LDD.

## 3.1 Background

**Markov decision process.** Consider a MDP $\mathcal{M} = \{\mathcal{S}, \mathcal{A}, P, r, \gamma\}$. Here, $\mathcal{S}$ and $\mathcal{A}$ respectively are the discrete state (e.g. language goals, descriptions, visual observations) and action spaces of the problem. $P(s_{t+1}|s_t, a_t)$ is the transition probability of transitioning into state $s_{t+1}$ by taking action $a_t$ from state $s_t$. $r(s, a)$ is the reward function given some state and action pair. $\gamma$ is a discount factor to prioritize short-term rewards.

**Actor-critic methods for policy learning.** In RL, we learn a policy $\pi(s; \theta)$ that maps from observations to actions $\pi : \mathcal{S} \to \mathcal{A}$. Let $\mathcal{R}(\tau)$ denote the total discounted reward over the trajectory $\tau$. The objective is to maximize the expected reward $J_\pi(\theta) = \mathbb{E}_\pi[\mathcal{R}(\tau)]$ following the policy $\pi$ by optimizing its parameters $\theta$. For trajectory length $T$, the policy gradient is

$$\nabla \mathbb{E}_\pi[\mathcal{R}(\tau)] \quad = \quad \mathbb{E}_\pi \left[ \left( \mathcal{R}(\tau) \sum_{t=1}^{T} \nabla \log \pi(a_t, s_t) \right) \right] = \mathbb{E}_\pi \left[ \left( \sum_{t=1}^{T} G_t \nabla \log \pi(a_t, s_t) \right) \right] \quad (1)$$

where $G_t = \sum_{k=0}^{\infty} \gamma^k r_{t+k+1}$ is the return or discounted future reward at time $t$. We consider the actor-critic family of policy gradient methods, where a critic is learned to reduce variance in the gradient estimate. Let $V(s) = \mathbb{E}_\pi[G_t|s_t = s]$ denote the state value, which corresponds to the expected returns by following the policy $\pi$ from a state $s$. Actor critic methods estimate the state value function by learning another parametrized function $V$ to bootstrap the estimation of the discounted return $G_t$. For instance, with one-step bootstrapping, we have $G_t \approx r_{t+1} + \gamma V(s_{t+1}; \phi)$. The critic objective is then $J_V(\phi) = \frac{1}{2} \left( r_{t+1} + \gamma V(s_{t+1}; \phi) - V(s_t; \phi) \right)^2$ We minimize a weighted sum of the policy objective and the critic objective $J_{ac}(\theta, \phi) = -J_\pi(\theta) + \alpha_V J_V(\phi)$.

## 3.2 Dynamics modeling during pretraining

In addition to policy learning, Language Dynamics Distillation learns a dynamics model from unlabeled demonstrations to initialize and distill into the policy learner. Consider a set of demonstrations without labeled actions $\mathcal{T}_\sigma = \{\tau_1, \tau_2, \ldots \tau_n\}$ obtained by rolling out some policy $\sigma(a_t, s_t)$, where each demonstration $\tau = [s_1, s_2, \ldots s_T]$ consists of a sequence of observations. We learn a dynamics model $\delta(s_1 \ldots s_t; \zeta)$ to predict the next observation $s_{t+1}$ given the previous observations.

$$J_\delta(\zeta) \quad = \quad \frac{1}{nT} \left( \sum_{i=1}^{n} \left( \sum_{t=1}^{T} sim \left( s_{t+1}, \delta(s_1, \ldots s_t; \zeta) \right) \right) \right) \quad (2)$$

where $sim$ is a differentiable similarity function between the predicted state $\delta(s_1, \ldots s_t)$ and the observed state $s_{t+1}$, and $\zeta$ are parameters of the dynamics model. In the environments we consider, $sim$ is the cross-entropy loss across a grid of symbols denoting entities present in the scene.

## 3.3 Dynamics distillation during policy learning

Fig 1 shows the decomposition of the model into a representation network $f_{rep}$, a policy head $f_\pi$, a value head $f_V$, and a dynamics head $f_\delta$. The three heads share parameters because their inputs are

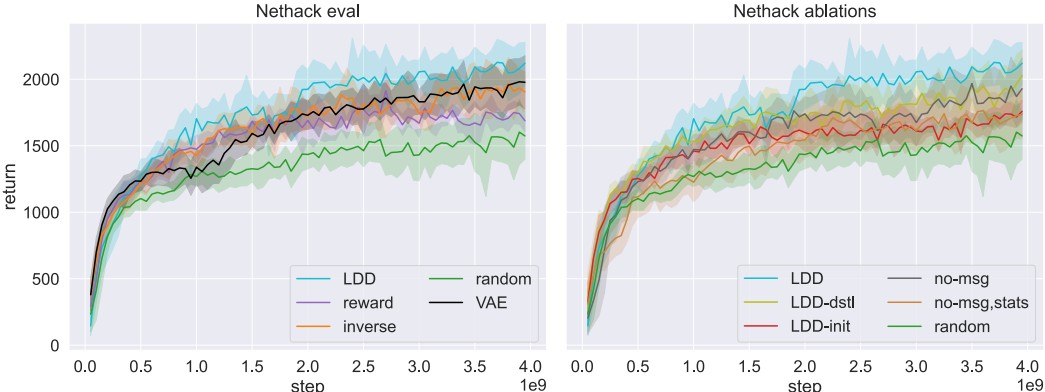

Figure 2: NetHack Challenge comparisons (left) and ablations (right). LDD consistently outperforms other methods.

formed by the same representation network.

$$\pi(s_t) = f_\pi \left( f_{\text{rep}} \left( s_t; \theta_{\text{rep}} \right); \theta_\pi \right) \tag{3}$$
$$V(s_t) = f_V \left( f_{\text{rep}} \left( s_t; \theta_{\text{rep}} \right); \theta_V \right) \tag{4}$$
$$\delta(s_t) = f_\delta \left( f_{\text{rep}} \left( s_t; \theta_{\text{rep}} \right); \theta_\delta \right) \tag{5}$$

In the first phase of dynamics modeling, we pretrain the model to predict future observations given demonstrations by optimizing $J_\delta$. We then store a copy of the model as a fixed teacher $\tilde{\delta}(s_t) = \tilde{f}_\delta \left( \tilde{f}_{\text{rep}} \left( s_t; \tilde{\theta}_{\text{rep}} \right) \right)$. Let $|X - Y|$ denote the L2 distance between $X$ and $Y$. During the second phase, in addition to policy learning, we optimize a distillation objective $J_d(\theta_{\text{rep}}, \theta_\pi, \theta_V) = |\tilde{f}_{\text{rep}} \left( s_t; \tilde{\theta}_{\text{rep}} \right) - f_{\text{rep}} \left( s_t; \theta_{\text{rep}} \right)|$ to avoid catastrophic forgetting of how to interpret language descriptions for dynamics modeling. This quantity is the similarity between the feature representation produced by the fixed teacher (e.g. the pretrained dynamics model) and the feature representation produced by the model. Because $\tilde{\delta}$ is frozen, the parameters $\tilde{\theta}_{\text{rep}}$ are not included in the objective function $J_d$. The joint loss for Language Dynamics Distillation is then

$$J(\theta_{\text{rep}}, \theta_\pi, \theta_V) = -J_\pi(\theta) + \alpha_V J_V(\phi) + \alpha_d J_d(\theta_{\text{rep}}, \theta_\pi, \theta_V) \tag{6}$$

To summarize, using unlabeled demonstrations with language descriptions, LDD learns a dynamics model of the environment that grounds language descriptions to environment observations (Section 3.2). This prior knowledge is then injected into reinforcement learning via initialization and distillation (Section 3.3).

## 4 Experiments

We evaluate Language Dynamics Distillation on the Situated Interactive Language Grounding benchmark (SILG) [Zhong et al., 2021]. SILG consists of five different language grounding environments with diverse challenges in term of complexity of observation space, action space, language, and reasoning procedure. In all environments, a situated agent observes symbolically (RTFM, Messenger, Nethack, Touchdown) or prose (ALFWorld) rendered visuals and interacts with the environment to follow some instance-specific language goals (e.g. what to do). In addition, the agent observes text manuals describing instance-agnostic environment rules (e.g. entity-role associations). The learning challenge is to learn a reading agent that generalizes to new environments with different environment rules (e.g. new entity-team associations, new parts of the map). The five different environments are as follows.

### 4.1 Environments

**NetHack** [Küttler et al., 2020]: The agent must descend a procedurally generated dungeon. Its primarily challenge is in large state space and partial observability, as the map remains obscured until

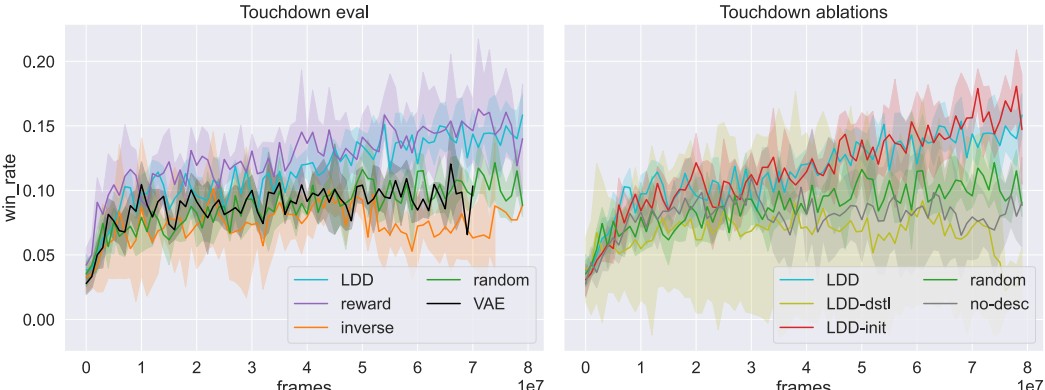

Figure 3: Touchdown comparisons (left) and ablations (right). Methods that distill or reward shape (LDD, `reward`, `LDD-init`) outperform those that do not.

exploration. Observations include a symbolic grid of entity IDs, in-game message, and description of character stats. The agent chooses among a fixed set of actions such as movement, picking up/buying/selling items, and attacking. We evaluate not on SILGNethack but the full NetHack challenge, a difficult game for humans with ∼15% expert win rate [Street, 2013].

**SymTouchdown** [Zhong et al., 2021]: A symbolic version of Touchdown [Chen et al., 2018] where the agent navigates segmentation maps of Street View panoramas following long instructions. The primary challenge is reading long, natural instructions that describe photorealistic images. Evaluation is on new navigation instructions. Observations include a grid of segmentation class IDs corresponding to a discretized Google Street View panorama, synthetic captions of where objects are relative to the agent (e.g. *to your right you see a lot of road and some cars*), and language navigation instructions. The agent selects from a list of radial directions to proceed to the next panorama.

**ALFWorld** [Shridhar et al., 2021]: The agent navigates and manipulates objects inside a kitchen which is described via textual descriptions. ALFWorld is challenging due to its large (>50) text action space that vary across scenes. Evaluation is on unseen instructions. Observations include a textual description of the scene and language goals (e.g. *put a clean sponge on the metal rack*). The agent chooses among a variable set of language actions (e.g. *open drawer 1*).

**RTFM** [Zhong et al., 2020]: The agent interprets a game manual and instruction to acquire the correct items to fight the correct monsters. Its main challenge is in multi-step reasoning that combines world observations with texts describing multiple entities. Evaluation is on a set of manuals distinct from those in training; hence the agent cannot memorize training manuals and must learn to read correctly in order to generalize. Observations include a symbolic grid containing names of entities present, manual describing high level game rules, agent inventory, and the language instruction. The agent chooses among a fixed set of movements. We train and evaluate on the first curriculum stage.

**Messenger** [Hanjie et al., 2021]: The agent delivers a message from a source entity to a target while avoiding an enemy. The entities are referred to in text by many names, which have no lexical overlap with their symbol ID, hence the core challenge is in mapping language entity references in text to observed symbolic entity IDs. Evaluations are on new entity-role assignments (e.g. who carries the message). Observations include a symbolic grid containing symbol IDs of entities present, and a manual of entities and roles. The agent chooses among a fixed set of movements. We train and evaluate on the second curriculum stage.

### 4.2 Method and Baselines

**Reinforcement learning with language descriptions from scratch.** We train a base tabula-rasa policy learner from random initialization. For NetHack, we train the base policy learner from [Küttler et al., 2020] using `moolib` [Mella et al., 2022]. For RTFM, ALFWorld, and Touchdown, we train the

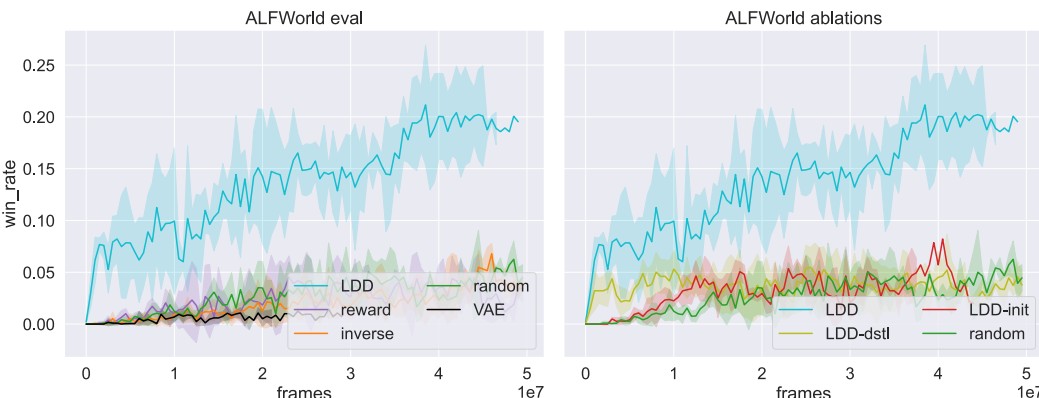

Figure 4: ALFWorld comparisons (left) and ablations (right). LDD consistently outperforms other methods.

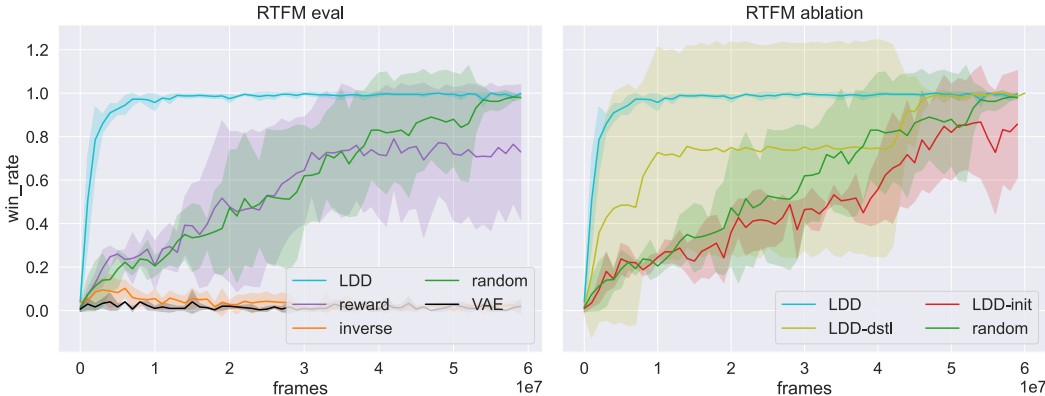

Figure 5: RTFM comparisons (left) and ablations (right). LDD consistently outperforms other methods. Because of the multi-step reasoning nature of RTFM solutions, partially complete strategies (e.g. able to do 2/3/4 cross-references) result in step-wise gains in win-rates. Strategies at different levels of completion result in larger variances when averaged.

SIR model from [Zhong et al., 2021] using Torchbeast [Küttler et al., 2019, Espeholt et al., 2018]. For Messenger, we train the EMMA model from Hanjie et al. [2021] using PPO [Schulman et al., 2017]. For NetHack, we train the base policy learner from [Küttler et al., 2020] using `moolib` [Mella et al., 2022]. For RTFM, ALFWorld, and Touchdown, we train the SIR model from [Zhong et al., 2021] using Torchbeast [Küttler et al., 2019, Espeholt et al., 2018]. For Messenger, we train the EMMA model from Hanjie et al. [2021] using PPO [Schulman et al., 2017].

**Pretraining representations via a variational autoencoder.** We pretrain a variational autoencoder (VAE), a common approach for representation learning, that predicts the intermediate representation just before the policy head [Kingma and Welling, 2013]. This VAE has the same architecture as the policy learner, and is used to initialize the policy learner. The training procedure for the VAE is as described in Ha and Schmidhuber [2018].

**Language Dynamics Distillation (**LDD**)** We train LDD variants of the baseline policy learners for each environment, where we pretrain the model to perform dynamics modeling on unannotated demonstrations. For NetHack, we use 100k screen-recordings (where actions are not annotated and cannot be trivially reverse engineered due to ambiguity in observations) of human-playthroughs from the alt.org NetHack public server. For Touchdown, we use unannotated demonstrations by human players. For ALFWorld, we use trajectories obtained from an A* planner with full state and goal knowledge, with actions removed. For RTFM and Messenger, we train expert trajectories until convergence, and sample 10k rollouts from the experts from which we remove action labels.

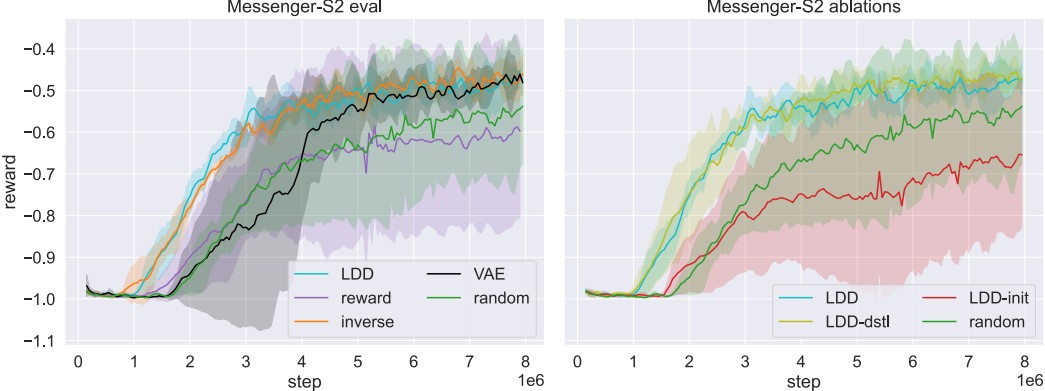

Figure 6: Messenger comparisons (left) and ablations (right). Methods that pretrain to initialize (LDD, `inverse`, `LDD-distill`) outperform those that do not.

**Reward shaping with expert.**    Methods such as Merel et al. [2017] reward shape with an expert by encouraging the agent to produce states similar to a demonstrator. To compare LDD to this idea, we use the dynamics model to predict the next observation under expert policy. The difference (e.g. accuracy in symbol prediction across grid) between the predicted observation and the actual observation after taking the action proposed by the agent is used as a penalty (e.g. negative auxiliary reward). This method is listing as `reward` in experiment figures and uses the same unlabeled demonstrations as LDD.

**Inverse reinforcement learning.**    Another class of methods learn a inverse dynamics model with which infer actions in unlabeled demonstrations, then learn to imitate the pseudo-labeled demonstrations [Hanna and Stone, 2017, Torabi et al., 2018]. To compare to this method, we first train the base policy learner for 10k episodes, then collect 10k rollouts to train an inverse dynamics model that predicts current action given current and future observations. This inverse model is used to annotate the original unlabeled demonstrations for imitation learning [Torabi et al., 2018]. Because these imitation policies do not generalize to novel environments and goals found in SILG evaluation, we additionally fine-tune them via RL similar to Guo et al. [2019]. This method is listing as `inverse` in experiment figures. In addition to the data used by `reward` and LDD, `inverse` uses additional rollouts to train the inverse dynamics model.

### 4.3   Results and ablations

LDD **consistently improves performance across environments.**    We evaluate on held-out environments across 4 random seeds for NetHack in Fig 2, Touchdown in Fig 3, ALFWorld in Fig 4, RTFM in Fig 5, and Messenger in Fig 6. LDD obtains top performance compared to tabula-rasa policy learning with language descriptions, VAE pretraining, reward shaping using the dynamics model, and inverse reinforcement learning. This is consistent across challenges in multi-step reasoning (RTFM), language-entity generalization (Messenger), large language action spaces (ALF-World), large procedurally generated states (NetHack), and long natural language instructions with complex visual scenes (Touchdown). We also ablate LDD by removing the initialization step (`LDD-init`) or the distillation step (`LDD-distill`). On Messenger, methods that pretrain to initialize (LDD, `inverse`, `LDD-distill`) outperform those that do not (`reward`, `LDD-init`). On Touchdown, methods that distill or reward shape (LDD, `reward`, `LDD-init`) outperform those that do not. Learning curves for each method across environments are shown in Appendix section G. LDD converges faster and to a higher win rate than other methods, with the exception of Touchdown, where it achieves lower training but higher evaluation win-rate.

**Adding language descriptions improves performance.**    What is the role of language descriptions in pretraining and subsequent policy learning? To answer this question, we ablate language descriptions by removing them from the environments in NetHack and SymTouchdown (the other environments are not solvable without descriptions because they describe the objective). `no-msg` removes NetHack messages describing events near the agent (e.g. *kitten attacks the bat!*). `no-msg,stats`

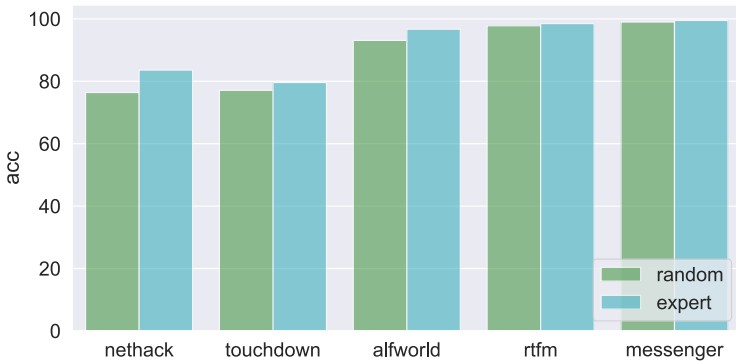

Figure 7: Dynamics model frame prediction accuracy using unlabeled expert demonstrations vs. rollouts from random policies. Training using expert demonstrations result in more accurate dynamics models, especially on complex environments that are difficult to explore via random policies.

additionally removes character state descriptions (e.g. health, achievements, dungeon level). `no-desc` removes Touchdown captions that describe objects locations scene relative to the agent (e.g. *on your left, there are many building, some people, and few cars*). These variants differ only in the observation space used in dynamics modeling. In fine-tuning, they receive the same observation space with language descriptions. For both NetHack (Fig 2) and Touchdown (Fig 3), removal of in-game messages, character state descriptions, and synthetic captions degrade performance. Appendix Fig 12 further shows that adding language descriptions results in dynamics models with higher accuracy. This suggests that modeling language descriptions in the observation space result in better initialization and distillation, which improve subsequent policy learning.

**Expert demonstrations cover late-stage strategy that result in asymptotic gains.** In complex environments, experts demonstrate late stage strategies difficult to explore via random sampling. Because these stages are rarely reached, accurate dynamics models are especially helpful in providing signals when rewards are sparse. Take NetHack as an example: unlike experts, non-expert policies never proceed to the deeper dungeons of the game. A dynamics model trained on non-expert rollouts therefore struggles to generalize to unseen deeper dungeons. As the agent learns and descends deeper in the dungeon, a dynamics model trained on non-expert demonstrations results in less distillation gains. Fig 7 compares dynamics modeling from observations using expert demonstrations vs. using rollouts from a random policy. The evaluation is on a held-out set of expert demonstrations. Across environments, training on expert demonstrations outperforms training on non-expert demonstrations. This effect is less apparent on environments where random policies can (eventually) discover most of the state space (e.g, RTFM, Messenger) and more apparent on partially observed environments where only strategic expert policies can encounter rare states indicative of success (e.g. long-term planning in NetHack and Touchdown, choosing from large language action space in ALFWorld).

## 5 Conclusion

While recent work showed that augmentation with language descriptions result in better policies, learning how to ground language descriptions to observations is difficult through naive RL with sparse, delayed rewards. We proposed Language Dynamics Distillation, which pretrains a dynamics model using cheaply-obtained unlabeled demonstrations with language descriptions to initialize and distill into the policy learner. On five tasks with language descriptions, LDD improved sample efficiency and resulted in better policies than RL from scratch, inverse RL, and expert reward shaping. In addition, the benefit from initialization and distillation differ on an environment basis, but are complementary across environments. Moreover, language descriptions improved initialization and distillation gains in policy learning. Finally, learning to model dynamics with expert demonstrations was more effective than with non-expert rollouts. A promising direction for future research is studying whether dynamics modeling with language descriptions is similarly effective in robotic control where naive RL can be prohibitively expensive, but unlabeled demonstrations with synthetic captions are cheap to obtain.

## Acknowledgments and Disclosure of Funding

We are grateful to the anonymous reviewers for their helpful comments and suggestions. Victor is supported in part by the ARO (AROW911NF-16-1-0121) and by the Apple AI/ML fellowship.

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
