## A Limitations

This work studies how language descriptions in unlabeled demonstrations benefit learning from observations. The environments used in this work are simulations. Despite variety across grounding challenges, performance on these environments do not necessarily transfer to other applications such as robotic control. A promising direction for future work is to investigate whether dynamics modelling on language observations show similar benefits in other applications.

## B Potential negative societal impacts

The methodology in this work are based on reinforcement learning, which may learn uninterpretable policies that achieve the objective in surprising ways (e.g. a robot that bumps along the cabinet while fetching dishes to clean). Language-conditioned policies are a way of controlling how policies behave by adjusting the language (e.g. instructions, in this case observations), however more research in this area is needed to develop methods that reliably understand and use language.

## C Code release

The source code for our experiments is available at

https://github.com/vzhong/language-dynamics-distillation.

## D Training details

We train and evaluate all methods on the SILG benchmark [Zhong et al., 2021], which comes with its own training and validation splits in terms of environment instances. We make distinction for Nethack, where we train and evaluate on the more difficult Nethack Challenge [Küttler et al., 2020], and Messenger, where we train and evaluate on the second curriculum stage as opposed to the first curriculum stage.

The code bases for each environments are based on the following work

1. Nethack: we use the https://github.com/facebookresearch/nle and its hyperparameters with the `human-monk` starting character. The demonstrations are 100k sampled `ttyrec` screen recordings downloaded from nethack.alt.org.

2. RTFM: we use the https://github.com/vzhong/silg and its default hyperparameters. The demonstrations are 10k sampled trajectories from a converged agent released with SILG.

3. Messenger: we use the https://github.com/ahjwang/messenger-emma and its default hyperparameters on the second curriculum stage. The demonstrations are 10k sampled trajectories from a converged agent trained using the default settings in EMMA on stage 1, then adapted to stage 2 via curriculum learning.

4. Touchdown: we use the https://github.com/vzhong/silg and its default hyperparameters. The demonstrations are the 6.5k human trajectories from the orinal Touchdown dataset [Chen et al., 2018].

5. ALFWorld: we use the https://github.com/vzhong/silg and its default hyperparameters. The demonstrations are the 21k full state planner trajectories from the original ALFRED dataset [Shridhar et al., 2020].

The dynamics models trained on these demonstrations are re-used for the reward-shaping method. To collect data for inverse dynamics modelling, we train a policy using the same hyperparameters for each environment for 10k episodes, then sample 10k episodes from the resulting policy. The sampled 10k episodes are used to learn a inverse dynamics model where two consecutive frames are used as the input and the inverse model predicts the action that took place between the frames. This inverse model is then used to predict actions on the demonstrations. The (pseudo-labeled) demonstrations are then used for imitation learning. The hyperparameters of the imitation learner is the same as those of the LDD experiments. This imitation learned model is then fined-tuned with RL.

Code for running the environment is anonymously submitted in the link in Section C. Hyperparameters for our experiments are obtained from Zhong et al. [2021] for RTFM, ALFWorld, and Touchdown; Hanjie et al. [2021] for Messenger, and Küttler et al. [2020] for NetHack. They are reproduced in Table 1 for convenience.

| Name | RTFM | Messenger | NetHack | ALFWorld | Touchdown |
|---|---|---|---|---|---|
| Base model | SIR | EMMA | ChaoticDwarf | SIR | SIR |
| Embedding size | 100 | 256 | 128 | 100 | 30 |
| RNN size | 200 | | 128 | 200 | 100 |
| Final repr size | 400 | 256 | 128 | 400 | 200 |
| Num FiLM$^2$ layers | 5 | | | 5 | 3 |
| Entropy cost | 0.05 | 0.05 | 0.001 | 0.05 | 0.05 |
| Baseline cost | 0.5 | 0.5 | 0.25 | 0.5 | 0.5 |
| Optimizer | RMSProp | Adam | Adam | RMSProp | RMSProp |
| Learning rate | 5e-5 | 1e-4 | 1e-4 | 5e-4 | 5e-4 |
| Optim epsilon | 0.01 | 1e-6 | 1e-6 | 0.01 | 0.01 |
| RMSProp alpha | 0.99 | | | 0.99 | 0.99 |
| Adam beta1 | | 0.99 | 0.99 | | |
| Adam beta2 | | 0.999 | 0.999 | | |
| Num actors | 30 | 30 | 128 | 30 | 8 |
| Learner batch size | 24 | 24 | 128 | 10 | 3 |
| Learner threads | 4 | 4 | 4 | 4 | 4 |
| Unroll length | 80 | | 64 | 80 | 64 |

Table 1: Hyperparameter settings. The base models SIR, EMMA, and ChaoticDwarf are respectively described in Zhong et al. [2021], Hanjie et al. [2021], and Küttler et al. [2020].

## E    Compute resources

We use a slurm cluster to train models. Each machine is equipped with a NVIDIA GPU with at least 16GB RAM and 20 CPU cores. Each run typically last 3 days, with the exception of ALFWorld (10 days) and Touchdown (6 days). Across 5 environments, we run 4 methods and 2 ablations for a total of 6 experiments. We additionally run 2 more language ablations experiments for Nethack and 1 more for Touchdown. Each experiment consists of 4 random seeds for a total of $(5 \times 6 + 3) \times 4 = 132$ runs. For the policy learning stage, our resource usage are on the order of $132 \times 10 \times 24 = 32k$ GPU hours or $132 \times 10 \times 24 \times 20 = 634k$ CPU hours. These experiments compose the bulk of our resource usage.

For dynamics pretraining, each run takes approximately 2 days of 1 GPU and 4 CPU. We re-use the trained dynamics model for reward shaping experiments. Inverse-dynamics modelling additionally require 1 day of pretraining an initial non-expert policy, generating rollouts from said policy, learning a inverse-dynamics model, and annotating unlabeled demonstrations with the inverse-dynamics model.

## F    Asset and license

We distribute this work under the MIT license. The dataset we use are publically available and distributed as a part of the SILG benchmark [Zhong et al., 2021]. There are no personally identifying information in the assets we use. SILG is distributed under a MIT license. The included environments are licensed as follows:

1. NetHack: NetHack General Public License
2. Touchdown: Creative Commons Attribution 4.0 International
3. ALFWorld: MIT
4. RTFM: Attribution-NonCommercial 4.0 International
5. Messenger: MIT

# G   Learning curves

Note that NetHack does not have a held-out evaluation set of environments.

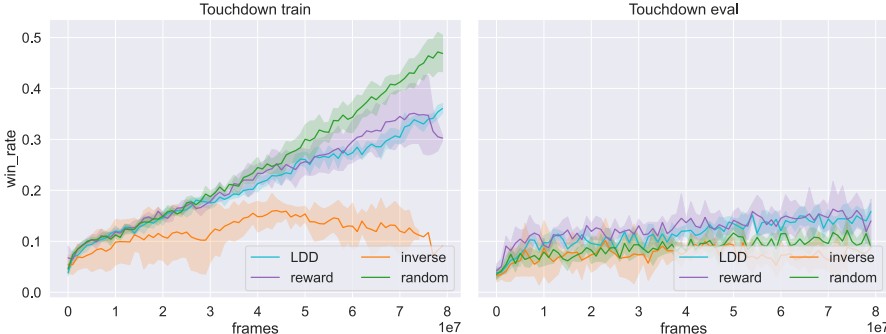

Figure 8: Touchdown learning curves

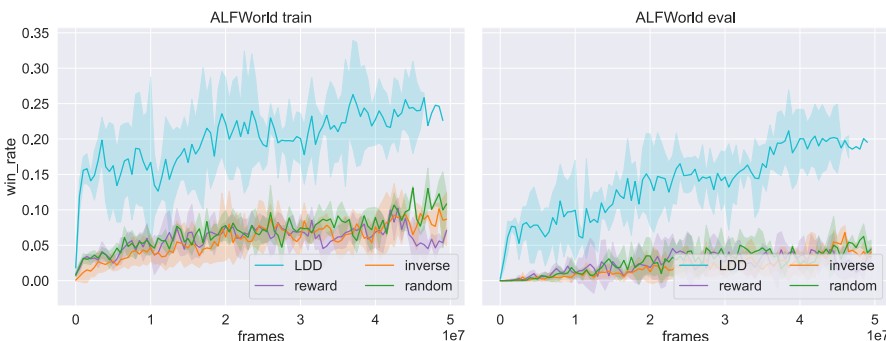

Figure 9: ALFWorld learning curves

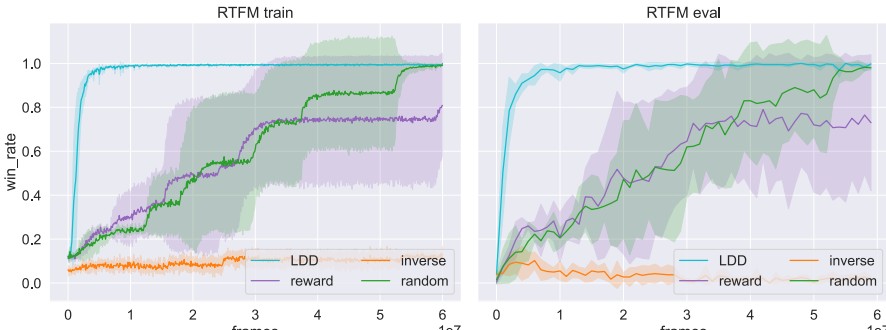

Figure 10: RTFM learning curves

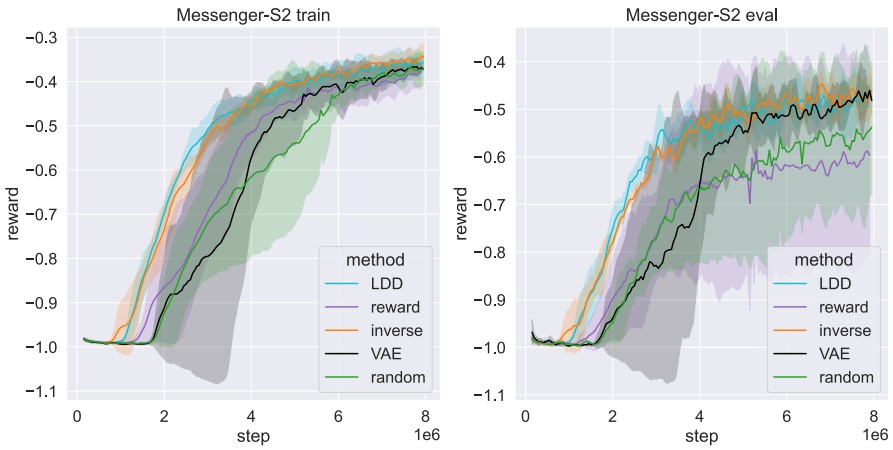

Figure 11: Messenger learning curves

## H  Dynamics modelling with vs. without language

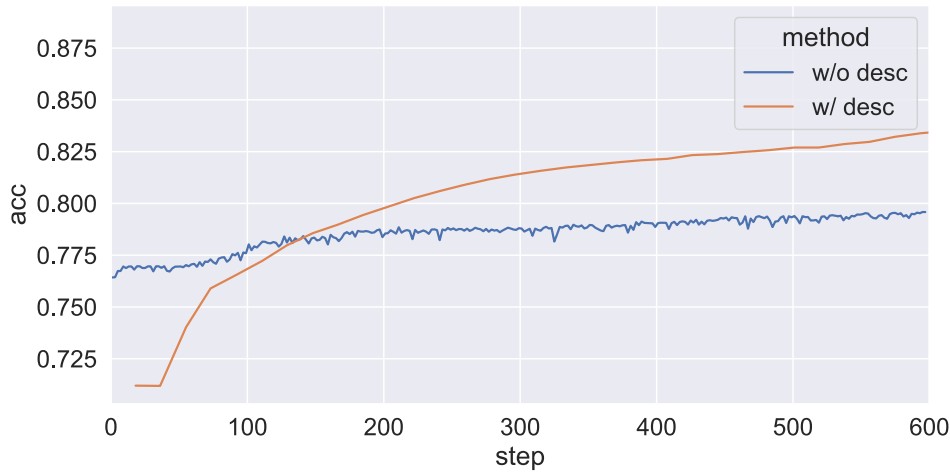

Figure 12: Dynamics modelling pixel-wise accuracy for SymTouchdown with vs. without language description inputs.