# OpenReview forum: "Improving Policy Learning via Language Dynamics Distillation"
_NeurIPS.cc/2022/Conference — NeurIPS 2022 Accept_

### Official Review · Reviewer_ZHyY · 2022-06-28

**Rating:** 6
**Confidence:** 4
**Soundness:** 3 good
**Presentation:** 4 excellent
**Contribution:** 3 good

**Summary:**

Sparse reward environments can be challenging for language-conditioned policies as credit assignment to the language description can prove difficult. The authors pretrain on a dataset of language-conditioned experts’ observations and distill this into the RL policy to demonstrate strong performance gains on a set of language-guided environments. I recommend accept for this paper, but would like to see a few things (mainly regarding ablations) addressed.

**Questions:**

None

**Strengths And Weaknesses:**

## Strengths

**Writing:** The motivation/method are presented clearly.

**Novelty:** The method is seemingly novel yet simple.

**Results:** LDD seems to work pretty well and provide strong improvements over the baselines in the paper across the some of the environments.

## Weaknesses

**Writing:** The last paragraph of the intro is unnecessarily long and reads more like an experiments analysis section. The intro would be more readable by greatly shortening this paragraph and leaving the detailed experiments results to the experiments section.

**Missing Ablation:** While I am pretty sure this baseline would be outperformed by LDD, the authors should test LDD trained without language guidance (just predicting next states). I see appendix Figure H does this partly, but this only compares with/without language inputs for dynamics modeling accuracy, not downstream RL performance.

Additionally, the authors should also include at least one other baseline that uses the same dataset in some way. For example, perhaps an LDD type of model that is pretrained with contrastive learning or some other representation learning objective (predict whether  observations are from the same trajectory given the language description), to verify that the dynamics loss is the best commonly used objective to apply here.

**Minor issues:**

- L220: “expert policy instead of for”
- Hyperparams: It’s useful for papers to at least have a small table/description of how hyperparameters were selected for all baselines and the main method, and how much tuning was done.

---

> ### Author Response · Authors · 2022-08-02
> **Author response**
>
> We thank the reviewer for their detailed feedback. We are glad that the reviewer finds motivations and methods clearly presented and that the method is novel yet simple. We are also glad that the reviewer agrees that the improvements LDD provides are significant. Please find below our point by point response, followed by a summary.
>
> ## Writing
> Thank you, we made the suggested changes.
>
> ## Ablation to show effect of language on downstream RL
> In the second paragraph of 4.3, we do have experiments showing that LDD performance drops when language is removed from observations during pretraining. In these two settings, the language descriptions are removed from unlabeled demonstrations. LDD is trained to predict just the observations as the reviewer suggested. As the reviewer hypothesized, this results in worse downstream RL performance on Touchdown and on NetHack. These are the only two environments where we can remove language guidance and still have the task remain solvable. For the other environments, removing language results in missing instructions or role assignments necessary for solving the task. We’re happy to discuss whether we should include this more aggressive ablation, but as the reviewer states, removing key language required to do the task would fundamentally limit the agent’s ability to generalize to new evaluation environments.
>
> ## Other baseline that use the dataset
> To clarify: the two comparison methods do use the same datasets. In reward shaping, the difference between the expert observations and agent observations is used as a negative reward. In inverse RL, an inverse model to label the unlabeled dataset for imitation learning. We thank the reviewer for catching this, and will clarify this section in the writing.
>
> ## Hyperparamters
> Thank you, we will add a table for the hyperparameters.
>
> In summary, we hope we have addressed the reviewer’s comments, specifically clarifying how the other two baselines use the same dataset of unlabeled demonstrations, and how our ablation experiment answers the question raised by the reviewer. We have made the suggested changes to the writing, as well as added a hyperparameter table to Appendix D. Given the positive qualities the reviewer highlighted, and our clarifications and changes, would the reviewer please look over the challenges, and if satisfied, consider increasing their score to 7?

---

> > ### Comment · Reviewer_ZHyY · 2022-08-03
> > **Response to authors**
> >
> > Thanks for the rebuttal response and making some of the proposed changes.
> >
> > I think you have addressed my concerns in writing. However, after reading the other reviews, I do agree with Reviewer W5e2 on having a representation learning comparison. The authors should compare against an observation representation learning method (or anything that doesn't focus on language) using the same dataset and also contextualize this paper with the representation learning for control field.
> >
> > This could, for example, be a VAE training objective for the representation model on reconstructing those discrete objects in the scene or a contrastive objective on observations being in the same trajectory or being augmented versions of the same data (like CURL).
> >
> > Furthermore, I think the authors should post a summary of changes made (and references to the new updated manuscript locations) that addresses all reviews, so that we all can easily see the concerns and corresponding changes the authors made in response to all reviews.
> >
> > I will consider changing my score if a proper representation learning baseline is included and some discussion on that is added; thanks again for the response.
> >
> >
> > Minor point:
> > One thing I noticed in the newest draft is that $\zeta$ isn't described in section 3.2, despite it being in the equation. The authors should clarify this.

---

> > > ### Author Response · Authors · 2022-08-03
> > > **Author response**
> > >
> > > Thank you for your prompt response!
> > >
> > > ## having a representation learning comparison.
> > > We will add an observation representation baseline that does not use language to this work.
> > >
> > > ## post a summary of changes made
> > > We have posted a summary of our changes as well as manuscript locations
> > >
> > > ## zeta isn't described
> > > We have defined $\zeta$
> > >
> > > Thanks again for your help!

---

> > > ### Author Response · Authors · 2022-08-07
> > > **Update with VAE results for Messsenger**
> > >
> > > As suggested, we implemented VAE pretraining and added its results for Messenger to Figure 6 and RTFM for Figure 5. The intermediate variable for VAE here being the representation just before the policy head. We found that VAE pretraining underperforms LDD (after considering the standard deviation). The other environments require substantially more time to run and we will not be able to finish the experiments by the end of the discussion period. Initial evidence from RTFM and Messenger suggests that LDD significantly outperforms VAE, especially when evaluation requires generalization to significantly different environment dynamics via reading (RTFM). Please let us know if you have any questions. Thank you again!

---

### Official Review · Reviewer_xVZE · 2022-07-11

**Rating:** 4
**Confidence:** 3
**Soundness:** 2 fair
**Presentation:** 1 poor
**Contribution:** 3 good

**Summary:**

This paper presents a new method, Language Dynamics Distillation(LDD), which pre-trains the model to predict environment dynamics with given demonstrations and language descriptions, then fine-tunes the final policy via reinforcement learning on pre-trained representations. The main contribution claimed by the authors is that this method can learn the RL policy more efficiently. It is especially effective when the reward is sparse or delayed, and it is a realistic setting because it assumes a cheap additional expert demonstration (ex. video) without action labeling.

**Questions:**

- I think the authors should have clearly defined and explained the problem setting (ex. about language description)  in Chapter 3. Does the defined s_t include a language description? If not, a detailed explanation of exactly how the language description is used seems necessary.
- Among the baseline algorithms, how was ‘reward shaping with expert’ (line 218-219) applied specifically?
- Can the proposed method be used with general intrinsic reward-related methods (ex. RND, ICM, …)? If so, I wonder if it still works effectively when applied with those methods.


**Limitations:**

The authors adequately addressed the limitations and potential negative societal impact of their work.

**Strengths And Weaknesses:**

Strengths
- It considers the realistic setting that assumes only cheap additional expert demonstration without action labeling.

Weaknesses
- It seems that the paper does not provide a clear explanation of the problem setting and the details of the proposed algorithm.
- It seems that the baseline algorithms provided in the experiment are not sufficient. It would be better if it was compared with several algorithms which are proposed to address the settings of sparse and delayed rewards. (ex. RND [1], ICM [2], …)

[1] Yuri Burda et al, Exploration by random network distillation, ICLR 2019

[2] Deepak Pathak et al, Curiosity-driven Exploration by Self-supervised Prediction, ICML 2017

---

> ### Author Response · Authors · 2022-08-02
> **Author response**
>
> We thank the reviewer for their detailed feedback. We are glad that the reviewer finds our problem setting realistic. Please find below our point by point response, followed by a summary.
>
> ## Clear explanation of the problem setting
> We initially left out the environment description to the SILG paper, but we will describe it in more detail in the paper. We thank the reviewer for helping us improve this work. For reference, the environments studied in our experiments involve a situated interactive agent which observes symbolically (RTFM, Messenger, Nethack, Touchdown) or prose (ALFWorld) rendered visuals and language goals for some game instances. In addition, the model observes text manuals describing instance-agnostic environment rules. The learning challenge is to learn a reading agent that generalizes to new environments with different environment rules (e.g. new entity-team associations, new parts of the map).
>
> ## Comparison to exploration methods
> LDD is not an exploration method. It can be used in conjunction with exploration methods such as RND (which is not applicable in positionless environments such as Touchdown, ALFWorld) and ICM. Follow on work may seek to experiment with the complementary gains provided by combining these methods, but this is out of scope for this paper.
>
> The distillation in LDD comes not from intrinsic predictiveness of how agent actions affect the environment, but from how agent observations differ from expert observations computed by the dynamics model. LDD emphasizes similarity to an expert whereas intrinsic exploration emphasizes novelty to prior experience, hence these methods are complementary. We thank you for bringing up this point of discussion and will aim to make this even clearer in the revised draft.
>
> ## Does s_t include language description?
> Yes it does. Thank you for raising this question. We will make this clear in the writing.
>
> ## How was reward shaping with expert applied?
> We train a dynamics model on unlabeled expert demonstrations. Each time the agent acts, we predict what should happen according to the expert dynamics model. We use the difference between the predicted observations and the observations from following the agent policy as a penalty to the agent. That is, the reward function encourages the agent to obtain observations similar to what an expert would obtain given the same state. The difference is taken as the mean position-wise symbolic ID accuracy, which may correspond to symbols, words, or object classes depending on the environment.
>
> In summary, we hope we have addressed the reviewers comments, specifically clarifying the exact problem setup and the differences between exploration methods and LDD. We also thank the reviewer for the clarification questions, which help improve the quality of this work. Our core contribution is a simple yet effective method to improve policy learning using unlabeled demonstrations. This method is complementary to exploration methods. We show that our method improves performances across seeds in five unique environments. We trust that the reviewer will agree that this is publishable given our clarifications and responses, and consider adjusting their score correspondingly.

---

### Official Review · Reviewer_cNeZ · 2022-07-13

**Rating:** 4
**Confidence:** 3
**Soundness:** 3 good
**Presentation:** 3 good
**Contribution:** 1 poor

**Summary:**

This paper proposes to ground policy representations on representation learned from language conditioned dynamic modeling. The paper argues that this is a cheap approach to obtain langauge grounding from observations, and illustrates across a set of different textual domains how such an approach can lead to improved RL performance.

**Questions:**

See above

**Limitations:**

Yes

**Strengths And Weaknesses:**

Strengths:

The paper is relatively clear to read, and the evaluation makes sense to me

Weaknesses:

The underlying proposed method seems a bit trivial to me -- the author simply proposed to finetune on a dynamics model.

The benefit of using expert data to train the dynamics model compared random data in Figure 7 seems to be minimal.

The underlying rewards curves of different models have very high variance. For example see the ablations in Figure 5.

The performance of the proposed method also appears to similar to other methods (see Figure 3)

Intuitively -- why does such a dynamic modeling objective lead to grounded language representations? Can the authors illustrate this more convincingly with more qualitative visualizations or quantitative grounding results? Also illustrating this qualitative difference with using random frames would be helpful.

---

> ### Author Response · Authors · 2022-08-02
> **Author response**
>
> We thank the reviewer for their detailed feedback. We are glad that the reviewer finds our work clearly written and our evaluations appropriate. Please find below our point by point response, followed by a summary.
>
> ## Just fine-tuning
> To clarify: in addition to initialization with the dynamics model, we also use it to distill during RL which we show improves results. We want to emphasize that the main contribution of this work is not to engineer a state-of-the-art method, but to answer the research question of how language, as a part of unlabeled demonstrations, can be used to improve learning.
>
> ## Simplicity
> We think that the fact that LDD is conceptually simple is a benefit. We ablated its components clearly across seeds on five environments and show that their combination improves results. Positive contributions in science and engineering do not have the requirement to be complex. Simple ideas that work are valuable, and many ideas seem simple once explained and demonstrated.
>
> ## Figure 7 gains
> The gains in Figure 7 should not be considered as minimal. In our draft, we note that gains in dynamics modelling performance are large for environments with sparse rewards such as NetHack and Touchdown. We see that the presentation requires clarification. We thank the reviewer for their help in improving this work.
>
> ## Figure 5 variance
> The variance in Figure 5 should not be considered as a weakness for any method. For some environments, the policy can get stuck on a local optimum. For example in Figure 5 RTFM (and Figure 6 Messenger), understanding monster assignments results in 25% win rate. Further understanding that items need to be obtained results in 50% win rate. Further obtaining the correct item results in 100% win rate. Because different seeds find these local optima at different times, averaging them results in large variance across time. We see that this needs to be clarified in our draft. We thank the reviewer for their help in improving this work.
>
> ## Significance of results
> Our results show that LDD consistently outperforms all methods on 4/5 environments across multiple seeds, with the exception of reward shaping on Touchdown. We argue that despite this single negative result, our experiments show that LDD is effective. We will enhance our writing to better scope our results.. We thank the reviewer for their help in improving this work.
>
> ## Qualitative analysis
> Our ablation results show quantitative differences between LDD and non LDD. We agree with the suggestion of adding additional qualitative analysis. We will examine the agent behaviour with and without LDD given the same environment state, and characterize the states when the policies differ. We are open to other qualitative analyses the reviewer suggests.
>
> In summary, we hope we have addressed the reviewers comments, specifically clarifying misunderstandings regarding distillation, variance, and the significance of results. We are also happy to discuss potential additional qualitative analysis. We hope that the reviewer considers increasing their score.

---

### Official Review · Reviewer_W5e2 · 2022-07-14

**Rating:** 6
**Confidence:** 4
**Soundness:** 3 good
**Presentation:** 3 good
**Contribution:** 2 fair

**Summary:**

The paper proposes to learn a language-conditioned dynamics model (without action labels), and shows that the representation learned by this dynamics model can improve downstream policy training, when combined with necessary techniques such as distillation. The method is evaluated on the SILG benchmark, which includes several distinct environments. Compared to baselines such as pure RL, inverse RL, reward-shaped RL, the proposed method is shown to outperform or on par with the best of all baselines across the benchmark. It is also ablated across several variants to show dynamics modeling with language outperforms without language, and that using expert trajectories for dynamics modeling is important.

--------------
Post Author Response Update

The authors have addressed my major concern for the paper. I'm updating my review score to 6.

**Questions:**

Please refer to the comments in the weakness section.

To summarize:

1. It is suggested to better contextualize the work around representation learning for RL, while advocating the importance of language modeling from this perspective.
2. In this regard, any comparison to a well-known or state-of-the-art method in “representation for RL” could be very useful to evaluate the significance of the contribution of this work.

**Limitations:**

The authors did not adequately address the limitations of this work; it is only stated that the limitation is that the environments are simulated. This is an insignificant limitation and not useful to evaluate the significance of the contribution of this work.

**Strengths And Weaknesses:**

Strengths:

1. The paper text is written with clarity.
2. The method appears to be novel for the reviewer.
3. The paper investigates important question in the field: how can language modeling help decision-making problems?
4. The evaluations include several distinct environments and tasks.

Weaknesses:

My concern is mostly around the significance of the paper. While it’s shown to be better than various baselines, the paper is not contextualized well enough for various prior works in representation learning for RL. While many prior works in this domain may not involve language modeling, given the nature of the proposed method, it should be discussed how it differs from other representation learning methods for RL and positions the paper from this perspective. Moreover, while the paper shows that the proposed method outperforms various baselines, it is susceptible that the improvement mostly comes from the good representation modeling. Therefore, this can only be a meaningful contribution if the paper could show comparisons to prior works in representation learning for RL. For example, what if the expert trajectories are used to train a representation module that is not a dynamics model, using some objectives? Would this representation work well in these tasks? If not, is it because they don’t model language?

Minor improvement include:

1. While the text is relatively clear in the paper, Figure 1 is poorly made, with blurry environment pictures and unclear method visualization. Potential improvement, for instance, can be separating the “traditional training loop” from the proposed method.
2. Fig 6b seems to be missing several learning curves.

---

> ### Author Response · Authors · 2022-08-02
> **Author response**
>
> We thank the reviewer for their detailed feedback. We are glad that the reviewer found our work clearly written and novel. We are also glad that the reviewer agrees that our work investigates an important question of how language can help decision making, and evaluates on a variety of environments and tasks. Please find below our point by point response, followed by a summary.
>
> ## Not contextualized well enough for various prior works in representation learning for RL
> We understand that a primary concern the reviewer has has to do with comparison to other approaches for representation learning for RL. However, the reviewer does not point out potential comparisons - what other techniques did the reviewers have in mind? We are happy to discuss potential experiments in detail.
>
> At a high level, it is difficult to disentangle language from representation learning for language grounding environments because language is necessary to solving the task. For example, the agent must understand the instructions and the language specifications in the manuals in order to figure out policies that generalize to new environments during evaluation. Consequently, we do compare LDD to representation learning for RL methods such as reward shaping, and inverse RL, because these methods all seek to learn better language-grounded representations. Moreover, for two environments where we can control for the presence of language, we show that describing demos with language improves performance (4.3 second paragraph).
>
> Finally, reward shaping with a dynamics model is a state-of-the-art technique for representation learning for RL. For example work concurrent with ours MineDojo: https://arxiv.org/abs/2206.08853, VPT: https://openai.com/blog/vpt. In our case, we lack millions of videos and annotated text, hence we rely on smaller dynamics models.
>
> We will add a detailed discussion of how to frame LDD within the larger landscape of representation learning for RL. Again, we are happy to discuss potential comparisons that the review has in mind.
>
> ## Figure 1 and 6
> We will update the figures in the paper. Thank you for pointing this out.
>
> ## Limitations
> Regarding the limitations sections - we’re happy to discuss how to improve our limitations sections: what improvements does the reviewer suggest?
>
> In summary, we hope we have addressed the reviewers comments, specifically with regards to how to frame LDD in work on representation learning for RL. We are also happy to discuss potential additional comparisons as well as improvements to the limitations section. We hope that the reviewer considers increasing their score.

---

> ### Author Response · Authors · 2022-08-07
> **Representation learning baseline**
>
> As suggested by ZHyY, we implemented a VAE pretraining baseline, a standard representation learning method for RL, and added its results for Messenger to Figure 6 and RTFM to Figure 5. The intermediate variable for VAE here being the representation just before the policy head. We found that VAE pretraining underperforms LDD (after considering the standard deviation). The other environments require substantially more time to run and we will not be able to finish the experiments by the end of the discussion period. We will add a detailed discussion of VAE results. Initial evidence from RTFM and Messenger suggests that LDD significantly outperforms VAE, especially when evaluation requires generalization to significantly different environment dynamics via reading (RTFM). Please let us know if you have any questions. Thank you again!

---

### Author Response · Authors · 2022-08-03
**Summary of changes**

We thank all reviewers for their detailed response. In summary, we are glad that the reviewers found our work clearly written (W5e2, cNeZ) and novel (W5e2, ZHyY), investigating an important question of how language can help decision making (W5e2), evaluates on a variety of environments and tasks (W5e2, cNeZ, xVZE), and makes significant improvements (ZHyY). Please find our point-by-point response in individual replies. Below is a list of major changes we have made to the draft as suggested by the initial round of reviews:

1. Added hyperparameter table to Appendix D
2. Modified Related works (section 2) paragraph 3 to frame LDD in prior work on representation learning
3. Modified Results (section 4.3) paragraph 3 to discuss results from removal of language
4. Added note to RTFM (Figure 5) explaining higher variance that result from averaging partially complete strategies
5. Added description of environment setup to Experiments (section 4) paragraph 1
6. Added a VAE baseline for representation learning. Initial results for Messenger is shown in Figure 6 and RTFM in Figure 5. Experiments for other environments are ongoing.

We have also revised to clarify several comments, such as explaining $s_t$, $\zeta$. We do not have time to run additional baselines during the response period because experiments take weeks to run, however we will add representation learning baselines (W5e2) and perform qualitative analyses (cNeZ) as suggested by the reviewers.

Thank you all again for your detailed feedback. Please engage with us during the discussion period as we strive to answer questions and clarify our presentation. We greatly value your help in improving this work!

---

### Meta-Review · Area_Chair_GEJc · 2022-08-26

**Recommendation:** Accept
**Confidence:** Less certain

**Metareview:**

This work proposes to learn better representations for language description-based tasks like navigation, via first pretraining a dynamics model on sequences of observations without action labels and using this model to aid RL-based policy learning. Good empirical results are presented and the work has been well-received. The dynamics policy learnt helps policy learning especially with longer horizons. The authors are encouraged to incorporate the reviewer feedback into account and especially the VAE experiments and add discussion.

**Award:**

No

---

### Decision · Program_Chairs · 2022-09-14

Accept